# Developing Gut-Healthy Strains for Pets: Probiotic Potential and Genomic Insights of Canine-Derived *Lactobacillus acidophilus* GLA09

**DOI:** 10.3390/microorganisms13020350

**Published:** 2025-02-06

**Authors:** Mengdi Zhao, Yuanyuan Zhang, Yueyao Li, Guangyu Li

**Affiliations:** 1College of Animal Science and Technology, Qingdao Agricultural University, Qingdao 266109, China; 20232109046@stu.qau.edu.cn (M.Z.); 20222103021@stu.qau.edu.cn (Y.Z.); 20222203016@stu.qau.edu.cn (Y.L.); 2College of Animal Science and Technology, Jilin Agriculture University, Changchun 130118, China

**Keywords:** canine, *Lactobacillus acidophilus*, whole genome sequencing, probiotic properties, safety assessment

## Abstract

Probiotics are widely used to improve pet health and welfare due to their significant biological activity and health benefits. *Lactobacillus acidophilus* GLA09 was derived from the intestinal tract of healthy beagles. The safety and suitability evaluation of GLA09 was completed through a combination of whole genome sequence and phenotypic analyses, including tests for the inhibition of harmful bacteria, acid resistance, bile salt tolerance, adhesion, and amine-producing substance content. The findings revealed that GLA09 has good gastrointestinal tolerance, inhibits the growth of pathogenic bacteria, and does not produce toxic biogenic amines. The genome of GLA09 comprises one chromosome and one plasmid, with a genome size of 2.10 M and a Guanine + Cytosine content of 38.71%. It encodes a total of 2208 genes, including 10 prophages, and 1 CRISPR sequence. Moreover, 56 carbohydrate-encoding genes were identified in the CAZy database, along with 11 genes for cold and heat stress tolerance, 5 genes for bile salt tolerance, 12 genes for acid tolerance, and 14 predicted antioxidant genes. Furthermore, GLA09 has one lincosamide resistance gene, but there is no risk of transfer. GLA09 harbors a cluster of Helveticin J and Enterolysin A genes linked to antimicrobial activity. Genomic analysis validated the probiotic attributes of GLA09, indicating its potential utility as a significant probiotic in the pet food industry. In summary, *L. acidophilus* GLA09 has the potential to be used as a probiotic in pet food and can effectively combat intestinal health in pets.

## 1. Introduction

*Lactobacillus acidophilus* is a microaerobic, Gram-positive bacterium classified within the phylum Firmicutes [1]. Since *L. acidophilus* was first isolated from infant feces in 1900, it has been recognized for its diverse probiotic effects on gastrointestinal health and overall well-being [2]. *L. acidophilus* can alleviate inflammatory bowel disease by reducing levels of inflammatory cytokines [3,4]. In addition, *L. acidophilus* serves multiple functional roles, including immune regulation, the enhancement of the intestinal epithelial barrier, the modulation of gut microbiota, cholesterol reduction, and the improvement of the body’s antioxidant capacity [5,6,7]. Recent studies have demonstrated that *L. acidophilus* can produce active metabolites such as extracellular polysaccharides and bacteriocins, which can directly or indirectly modulate host metabolism and immune responses [8,9]. Although numerous studies have documented the biological functions and probiotic properties of various *L. acidophilus* strains, most research has focused on physiological aspects. In contrast, relatively few studies have been conducted on the genetic function of *L. acidophilus*.

With the advancement of high-throughput sequencing technology, many studies have been conducted to analyze the safety and probiotic properties of strains at the genomic level, such as *Lacticaseibacillus rhamnosus*, *Limosilactobacillus reuteri*, *Lactobacillus johnsonii*, *Bifidobacterium longum* [10,11,12,13]. Whole-genome sequencing (WGS) reveals gene function, metabolic pathways, genetic features, and the genome structure of microorganisms [14]. WGS, along with bioinformatics analysis, has emerged as a crucial methodology. Microorganisms derived from the host are preferred as probiotics over non-host microorganisms because they have co-evolved with their hosts for approximately 10 million years, are better adapted to the host’s gastrointestinal environment, exhibit superior adherence, and provide lasting effects [15,16,17]. Currently, while some studies on probiotics in dogs have been conducted, there remains a scarcity of research specifically focusing on canines, prompting researchers to seek improved probiotic options for the future [18]. In this study, *L. acidophilus* GLA09 was isolated from the gastrointestinal tract of healthy beagle dogs. The objective was to investigate the potential probiotic effects of GLA09 as a pet food probiotic and to evaluate its safety through whole genome sequencing (WGS), mass spectrometry detection, gene sequence prediction, and functional annotation, thereby providing a theoretical basis for its application.

## 2. Materials and Methods

### 2.1. Main Reagents and Instruments

MRS broth medium (MRS, Haibo, Qingdao, China); Luria-Bertani (LB, Haibo, Qingdao, China); DNA extraction kit (Solarbio, Beijing, China); sequencing platform: NovaSeq 6000 (Illumina, San Diego, CA, USA).

### 2.2. Strain Source and Culture

*L. acidophilus* GLA09, derived from canine sources, was isolated from the gastrointestinal tract of healthy beagles in our laboratory and preserved at the Chinese Typical Cultures Depository Center in Wuhan, China, under the deposit number CCTCC NO: M2023982. In this study, *L. acidophilus* GLA09 was cultured in a 10 mL MRS broth at 37 °C with shaking for 24 h to activate the strain. Subsequently, it was passaged three times with a 3.0% (*v*/*v*) inoculum before being prepared for use.

The strains were inoculated into MRS liquid medium at an inoculum of 3.0% and incubated at 37 °C for 48 h. The control was a blank MRS liquid medium without strains. During the incubation period, OD600 nm absorbance was measured at 0, 1, 2, 3, 6, 9, 12, 15, 18, 21, 24, 36, and 48 h. The growth curves were plotted with time as the horizontal coordinate and OD600 absorbance as the vertical coordinate (subtracting the OD600 nm of the control group, removing the effect of the base color of the medium).

### 2.3. WGS, Assembly and Annotation

#### 2.3.1. Strain DNA Extraction and Detection

After a 24-h incubation period, the bacterial solution was centrifuged at 4 °C at 10,621× *g* for 10 min, and the supernatant was discarded to collect the bacterial pellet. Bacterial DNA was then extracted using a bacterial DNA extraction kit (Qiagen, Beijing, China), followed by a quality assessment of the DNA concentration and purity using an ultra-micro spectrophotometer and agarose gel electrophoresis.

#### 2.3.2. WGS and Genome Assembly

The WGS analysis was conducted using the Oxford PromethION platform and the Illumina NovaSeq 6000 platform. The genome was assembled de novo from the filtered reads using Unicycler software (version 0.5.0) based on the sequencing data. Subsequently, Pilon version 1.24 was employed to further refine the assembled genome using second-generation data [19].

#### 2.3.3. Functional Annotation of Genomes

The coding genes were predicted from the assembled genome using Prokka software (version 1.14.6) to obtain general genomic features. To gather more comprehensive information on gene function, gene function annotation was performed using BLAST+ (version 2.11.0+) via Gene Ontology (GO), Kyoto Encyclopaedia of Genes and Genomes (KEGGs), and Cluster of Orthologous Groups of Proteins (COGs) (version 2.11.0+) comparisons [20,21,22]. A carbohydrase annotation was performed according to the Carbohydrate-Active EnZymes database (CAZy, https://www.cazy.org, accessed on 20 July 2023), and strain safety prediction was conducted using the Pathogen Host Interactions Database (PHI version 4.12, http://www.phi-base.org/, accessed on 20 July 2023) (identity > 40%), Comprehensive Antibiotic Research Database (CARD version 3.2.0, http://arpcard.mcmaster.ca, accessed on 20 July 2023), Pathogenfinder 1.1 (https://cge.food.dtu.dk/services/PathogenFinder/, accessed on 20 July 2023), and Virulence Factor Database (VFDB, http://www.mgc.ac.cn/VFs/, accessed on 20 July 2023, identity > 60%) [23,24,25].

#### 2.3.4. Secondary Metabolites

The predictions of secondary metabolites were performed using antiSMASH 7.0 (https://docs.antismash.secondarymetabolites.org/, accessed on 20 July 2024).

#### 2.3.5. Comparative Genomes

Genome evolutionary trees were constructed using a Type Strain Genome Server (https://tygs.dsmz.de, accessed on 26 June 2024), and comparative genome analysis was performed using TBools v2.056.

### 2.4. Biogenic Amines Detection

The content of biogenic amines produced by GLA09 was analyzed using LC-MS/MS (Sciex 4500, Shanghai, China). A 100 μL sample was taken, and 100 μL of sodium bicarbonate solution, 10 μL of sodium hydroxide solution, and 100 μL of dansyl chloride solution were added. The mixture was vortexed and mixed thoroughly. Subsequently, it was incubated in a 60 °C water bath for 25 min, cooled to room temperature, and 10 μL of ammonia was added. The sample was then concentrated to dryness and redissolved in 1 mL of acetonitrile. The supernatant was filtered through a 0.22 μm membrane, and the final sample volume was 3 μL. The chromatography column used was a HYPERSIL GOLD C18 column (3 μm, 2.1 mm × 100 mm) with a column temperature of 35 °C and a flow rate of 0.3 mL/min. The criteria for determining Biogenic amines were referenced from Lorencová et al. [26].

### 2.5. In Vitro Probiotic Potential of Strain

#### 2.5.1. Cell Surface Hydrophobicity and Auto-Aggregation

The auto-aggregation assays referred to the method by Zhao et al. [27]. Following a 24-h growth period, the isolated cultures were washed three times with phosphate-buffered saline (PBS, pH 7.0) and adjusted to an approximate concentration of 1 × 10^8^ CFU/mL. The bacterial suspension was vortexed for 30 s, and the initial absorbance was measured at OD_600_ nm (*A*_0_). Subsequently, the solution was incubated at 37 °C for 2 h. The OD value (*A*_1_) was determined at 600 nm using PBS buffer as a control. The aggregation activity was then calculated using the following equation: (1)$$\text{Auto-aggregation}\;\mathrm{activity}\;(\mathrm{AAG},\%)=(1-A_{1}/A_{0})\times 100\%$$

The cell surface hydrophobicity assays referred to the method by Chen et al. [28]. GLA09 was cultured for 24 h, then washed three times with PBS (pH 7.0), and adjusted to a concentration of approximately 1 × 10^8^ CFU/mL. The absorbance at OD_600_ nm was measured (*A*_2_). An equal volume of xylene and chloroform was then added, and the mixture was vortexed for 30 s before being incubated at 37 °C for 1 h. The organic phase was removed, and the OD value of the aqueous phase was determined at 600 nm (*A*_3_). The cell surface hydrophobicity was then calculated using the following equation:(2)$$Cell\;surface\;hydrophobicity\;(CSH,\%)=(1-A_{3}/A_{2})\times 100\%$$

#### 2.5.2. Antibacterial Activity

The antibacterial activity of GLA09 was detected using the Oxford cup double-layer agar diffusion method [29] with slight modifications. Following incubation, the isolates were harvested from an MRS liquid medium and adjusted to a concentration of approximately 1 × 10^7^ CFU/mL. Moreover, the pathogenic bacteria (*Escherichia coli* ATCC 25922, *Staphylococcus aureus* ATCC 25923, and *Salmonella typhimurium* ATCC 14028) were cultured in an LB liquid medium and adjusted to 1 × 10^7^ CFU/mL. The plates were then incubated at 37 °C for 24 h, after which the inhibition zone diameter (IZD) was measured. A blank MRS medium was used as a negative control and laboratory strain CLP03 as a positive control (previously tested for bacteriostatic properties) [27].

#### 2.5.3. Low pH and Bile Salt Tolerance

Additionally, the growth curve of GLA09 at different pH and bile concentrations was analyzed. To enhance the experimental procedure described by Ramos et al. [30] for determining the acid and bile salt tolerance of strains, bacteria were inoculated into modified pH-MRS (pH = 1.5–4.0) and BS-MRS (MRS medium with bile salt concentrations of 0.1%, 0.2%, and 0.3%) with an inoculum size of 3.0%, and samples were collected at 0, 2, 4, and 8 h. All experiments were conducted in triplicate for each experiment.

#### 2.5.4. Data Analysis

Statistical analysis of the data in vitro probiotic potential was expressed as the mean ± standard deviation (SD). Each experiment was repeated three times independently.

## 3. Results

### 3.1. Genomic Characterization of L. acidophilus GLA09

By WGS of *L. acidophilus* GLA09, 9,487,074 valid reads remained after filtering. The size of the assembled complete genome was 2,102,723 bp, with a Guanine + Cytosine (G + C) content of 38.71%. As shown in Figure 1A, it contained one chromosome and one plasmid with sizes of 2,042,740 bp and 59,983 bp and G + C contents of 34.94% and 33.38%, respectively. The total number of coding genes was 2208, and the total length of coding genes was 1,860,808 bp, accounting for 88.50% of the total genome length, with an average gene length of 843 bp. In addition, the non-coding genome of the whole genome of strain GLA09 contained 64 tRNAs, 5 23S rRNAs, 5 16S rRNAs, 5 5S rRNAs, eight gene islands, 10 prophages, and one CRISPR sequence was predicted (Table 1 and Appendix A). Furthermore, the results of strain GLA09 alignment were similar to those of *L. acidophilus* NBRC 13951 using the Type genome server (Figure 1B).

### 3.2. Functional Annotation Analysis of the L. acidophilus GLA09

Prokka is a tool designed for rapid annotation of prokaryotic genomes [31]. The genome-wide coding genes of strain GLA09 were annotated using universal database annotation; the statistical results are shown in Figure 2 and Figure 3. A total of 2076 functional genes were annotated, of which the number of genes that appeared in at least one database was 2062, which accounted for 99.33%.

#### 3.2.1. COG Annotations

The COG database is a database developed by the NCBI for the annotation of homologous proteins. The strength of the COG database lies in the accuracy and comprehensiveness of its classification. There are 2005 genes in the whole genome of *L. acidophilus* GLA09 completed protein annotation. The highest number of annotations was for translation, ribosome structure, and biosynthesis with 183 genes, followed by carbohydrate transport and metabolism with 174 genes, major function prediction with 167 genes, amino acid transport metabolism annotated to 160 genes, and transcription annotated to 149 functional genes (Figure 2B).

#### 3.2.2. GO Annotations

GO is a database created by the Gene Ontology Consortium that provides a comprehensive description of the properties of genes and gene products in organisms. A total of 1120 genes were annotated, and the percentage of genes with GO annotation was 53.95% (Figure 2C). Among the three major functional gene categories of GO, the top three functional annotations in biological processes were translation, transposition, DNA-mediated, and DNA integration; the more annotated genes in cellular composition were integrated. Among the cellular components, the most frequently annotated are an integral component of the membrane, cytoplasm, plasma membrane, and ribosome. Among the molecular functions, the most frequently annotated were binding, ATP binding, and DNA binding.

#### 3.2.3. KEGG Annotations

The KEGG database is a systematic analysis of the metabolic pathways of gene products in cells and the functions of these gene products. A total of 1203 functional genes were annotated by KEGG for *L. acidophilus* GLA09 (Figure 3A). Metabolism had the highest proportion, while cellular processes had the lowest proportion. Among the metabolic processes, 128 functional genes were related to carbohydrate metabolism, 76 genes were associated with amino acid metabolism, and 61 genes were related to nucleotide metabolism. In addition, there were 124 genes in the environmental information processing process related to membrane transport.

#### 3.2.4. TCDB Functional Notes and Pathogen Host Interactions

PHI annotation results show 215 Unaffected pathogenicity and 484 Reduced virulence (Figure 3B). Furthermore, a total of 421 membrane transporter protein-coding genes were annotated in the TCDB database (Figure 3C). Among them, the major active transporter proteins were annotated with 184 coding genes, followed by electrochemical potential-driven transporters with 94 coding genes.

#### 3.2.5. Non-Redundant Protein Database

*L. acidophilus* GLA09 using the Non-Redundant Protein database, the results show that strain GLA09 with *L. acidophilus* had the highest rate, at 80.04% (Figure 3D).

#### 3.2.6. CAZy Functional Notes

*L. acidophilus* GLA09 was annotated with a total of 56 carbohydrate-encoding genes in the CAZy database (Figure 3E). Glycoside hydrolase genes were annotated with 34, with the highest percentage of 60.71%, followed by glycosyltransferase genes with 19, with 33.93%; a carbohydrate-binding module was annotated with 2 genes, representing 3.57%, and carbohydrate lipase was annotated with only 1 gene.

### 3.3. L. acidophilus GLA09 Tolerance-Related Genes

*L. acidophilus* GLA09 annotated to 3 stress protein genes, 8 heat tolerance-related genes, 3 cold tolerance-related genes, 5 bile salt tolerance-related genes, 12 acid tolerance-related genes, and 14 antioxidant-related genes (Table 2).

### 3.4. L. acidophilus GLA09 Safety-Related Gene

Annotation using the CARD database was performed for *L. acidophilus* GLA09 with one lincosamide resistance gene (Table 3). Moreover, the number of virulence factors annotated to 8 with a complete match rate greater than 60% (Table 4).

### 3.5. Biogenic Amine Content of L. acidophilus GLA09

*L. acidophilus* GLA09 was not detected in the supernatant of biogenic amine, showing that the strain is safe (Appendix A).

### 3.6. Secondary Metabolite Prediction and Comparative Genome

*L. acidophilus* GLA09 produced three bacteriocins as predicted in Figure 4, along with a comparative genome analysis (Appendix A). The first one was Bacteriocin Helveticin J, with a match of 99.61%, which was in the genome between 498,704~519,466 bp. The second one was Enterolysin A, with a match of 58.39%, which was in the genome. And, the third one is Helveticin J, with a matching degree of 44.66%, which all belong to the class III bacteriocins.

### 3.7. Results of In Vitro Probiotic Potential Test of the Strain

*L. acidophilus* GLA09 exhibits an excellent bacteriostatic effect (Appendix A) and superior adhesion. Moreover, it can withstand pH levels of ≥2.5 and is resistant to 0.3% (Figure 5).

## 4. Discussion

In this study, the whole genome of *L. acidophilus* GLA09 of canine origin was sequenced and mapped. The genome size of *L. acidophilus* GLA09 was 2.103 M, containing one chromosome and one plasmid, which belonged to the medium-sized genome. The G + C content was 38.71%, slightly higher than the average G+C content of *L. acidophilus* of 34.66% [1]. In this study, the strains were functionally annotated using GO, KEGG, and COG databases. The most functionally annotated genes in the GO database were integral components of the membrane, and the proteins encoded by these genes may have transmembrane transport abilities. Another significant annotation was ATP binding, which hydrolyzes ATP to provide energy for other biological processes. KEGG is a database that analyzes the metabolic pathways of gene products in cells and their functions of gene products at the molecular level [22]. *L. acidophilus* GLA09 had the most annotations in the KEGG database for metabolism-related genes. Studies have shown that *L. acidophilus* can produce large amounts of organic acids such as lactic acid, acetic acid, and butyric acid during fermentation. Additionally, it can metabolize and produce bacteriocins, extracellular polysaccharides, and other substances [32,33,34]. These metabolites contribute to the productivity of *L. acidophilus* and are utilized in various functional product developments, such as preventing diarrhea, lowering cholesterol, improving immunity, and other health-related products. Furthermore, the strains were determined to contain CRISPR sequences. The CRISPR/Cas system has been extensively utilized for genome editing. In the future, the development of efficient and precise genome editing tools is anticipated to enhance the research and application of gene functions [35].

Carbohydrates play a crucial role in many biological functions, and the CAZy database focuses on analyzing genomic, structural, and biochemical information on carbohydrate enzymes [36]. In this study, the highest number of glycoside hydrolase genes were annotated, demonstrating that glycoside hydrolases promote the hydrolysis of α-1,4 glycosidic bonds present in carbohydrates such as starch and also play a significant role in industrial production [37,38]. This study highlights that glycoside hydrolases can facilitate the hydrolysis of α-1,4 glycosidic bonds in carbohydrates such as starch, and are crucial for industrial production. Therefore, it is suggested that *L. acidophilus* GLA09 may enhance the digestive utilization of dietary carbohydrates or regulate glucose metabolism in the body.

*L. acidophilus* GLA09 genomic analysis revealed the presence of genes associated with resistance to cold and heat shock stress, bile salts, pH stress, and antioxidant stress. Firstly, concerning hot and cold stress, *L. acidophilus* is frequently exposed to high- or low-temperature extremes in industrial production. For example, when used as feed additives, high temperatures are generated during granulation and mixing or when fermented milk is subjected to prolonged low-temperature stress during cold storage [39,40]. The stresses of heat and cold can easily lead to the development of the bacterium. These hot and cold stresses can easily lead to sub-lethal or lethal conditions in the strains, thus reducing the probiotic function of the probiotics. In this study, *L. acidophilus* GLA09 was predicted to possess eight heat stress-related genes and three cold stress-related genes, suggesting that L. acidophilus GLA09 may exhibit better resistance to hot and cold stress, making it suitable for industrial applications. Secondly, probiotic bacteria can withstand low pH and high bile salt concentrations in the gastrointestinal tract, colonizing the intestine for long-lasting probiotic effects [41]. *L. acidophilus* GLA09 was predicted to have several genes related to pH tolerance and bile salt tolerance. Among them, the Na^+^/H^+^ ion reverse transporter protein and F0F1-ATP synthase have been shown to maintain intracellular pH homeostasis and pump H+ out of the cell to help the strain defend itself against the acidic environment [42]. The bile salt hydrolase gene may help to improve bile salt tolerance and increase the number of strains surviving in the gastrointestinal tract. The primary requirement for probiotics to be effective is their ability to withstand the harsh conditions of the stomach. GLA09 possesses multiple genes associated with resistance to pH stress and bile salts. This finding aligns with an in vitro study that demonstrated the strain’s capacity to tolerate pH levels of 3.0 and a wide range of bile salt concentrations.

In addition, oxidation is a necessary process for cellular metabolism in organisms, and the high levels of oxidative stress produced when organisms are subjected to oxidative stress can lead to aging as well as cause various chronic diseases [43]. *L. acidophilus* GLA has been reported as a natural antioxidant [44,45]. Several antioxidant-related genes were predicted in the whole genome of *L. acidophilus* GLA09. Therefore, it was hypothesized that *L. acidophilus* GLA09 might improve the antioxidant capacity of the body by increasing the activity of oxidative enzymes and so on. Additionally, three class III bacteriocins were predicted by the secondary metabolite database, containing two Helveticin J genes and one Enterolysin A gene. Bacteriocins produced by lactic acid bacteria are usually bacteriostatic, and Helveticin J bacteriocin is a chromosomally encoded bacteriocin with narrow-spectrum bacteriostatic effects. Enterolysin A bacteriocin has broader bacteriostatic effects by breaking down the cell walls of sensitive bacteria, including Streptococcus, Listeria monocytogenes, and *Lactococcus lactis* [46,47]. Therefore, it is suggested that *L. acidophilus* GLA09 may likewise be bacteriostatic and thus have the effect of regulating the intestinal flora.

The safety assessment of strains is another important consideration in the selection of potential probiotics. Although one lincosamide resistance gene was predicted, there was no risk of metastasis. Antibiotic resistance can occur through the horizontal transfer of antibiotic-resistance genes [48]. Mobile genetic elements, such as transposon-promoted enzyme expression, integrase, or recombinase, contribute to the initial mobilization. Mobile genetic elements are able to capture antibiotic resistance genes from chromosomes and transfer them horizontally to other bacteria via plasmids or phages [49,50]. In this study, strain GLA09 was predicted to have one lincosamide resistance gene by the CARD database, but it was located on the chromosome and had no repetitive sequences before or after it to constitute a transposon [51]. It was also not located on mobile genetic elements such as phages and gene islands as predicted by the genome, so it did not have the possibility of transfer and was not at risk of antibiotic transfer [52].

The virulence factors are divided into two categories: surface factors involved in host cell colonization and factors that lead to host tissue damage [53]. Classification is ambiguous in probiotics, and a large number of genes found in the VRDB are associated with virulence factors in some pathogens but with adaptive factors in probiotics [52]. The VFBD database is dedicated to collecting information on the virulence factors of bacterial pathogens. There are eight virulence genes with greater than 60% genetic similarity predicted in the whole genome of *L. acidophilus* GLA09. Of these, *EF-Tu* and *GroEL* are adhesion-associated factors that promote the adhesion and colonization of strains within the stomach [54]. The two-component response regulator *LisR* is ubiquitous in bacteria, facilitating their adaptation to diverse environments and playing a critical role in the regulation of oxidative stress [55]. In addition, the GLA09 genome is predicted to have a UTP, glucose-1-phosphate uridylyltransferase *GalU*, which enhances the resistance of the strain to freeze-drying [56]. There is a fraction of pathogenic virulence factors, but there is also a fraction of virulence factors that are not involved in pathogenicity but are essential for probiotics and are health factors. Therefore, whether these virulence genes are expressed in *L. acidophilus* GLA09 needs to be further verified. The PHI database is currently commonly used for the detection of pathogenicity genes, virulence genes, and effector protein genes in bacteria. The *L. acidophilus* GLA09 genome had the highest number of virulence-reducing genes annotated at 484, followed by 215 genes annotated with no effect on pathogenicity. The presence of many virulence-reducing and non-pathogenicity genes suggests that *L. acidophilus* GLA09 may have a pathogenicity-suppressing effect. Acceptable reference limits for histamine and tyramine are 0–100 and 100–800 mg/Kg [57,58]. For other amines, no criteria have been established in the literature. According to the Chinese national standard GB/T 5009.208-2016 [59], the minimum limit of BAs is 2–5 ug/mL. In this study, strain GLA09 produced less than 2 ug/mL of biogenic amines in all cases, and the strain is safe [60,61].

Although this study effectively established a connection between the genome and the probiotic properties, further validation of metabolites (bacteriocins), native animal testing, and assessments of product industrialization are necessary to provide substantial technical support for the strain as a probiotic product for pets. To ensure the quality of our probiotic product, GLA09, it is essential to establish standardized production processes. This entails optimizing culture conditions, as well as handling and storage practices, to guarantee the product’s effectiveness and stability.

## 5. Conclusions

In this study, the whole genome of *L. acidophilus* GLA09 was sequenced, and its genes were functionally annotated to analyze the probiotic properties and safety of the strain at the genetic level. The genome mainly related to functional genes includes carbohydrate metabolism, amino acid metabolism, nucleotide metabolism, and three predicted genes coding for bacteriocins. Additionally, a significant number of probiotic-related genes were identified in the *L. acidophilus GLA09* genome, including *trxA*, *mvaA*, *trxB*, *TRR*, *pcaC*, *GSR*, *gor*, and *fabG*. These genes are closely associated with antioxidant functions and contribute to the unique genomic features of this strain. The safety of the strain was elucidated at the gene level. Although one lincosamide resistance gene was predicted, there was no risk of metastasis. Therefore, *L. acidophilus* GLA09 of canine origin has excellent probiotic properties and safety and can be a candidate strain for probiotics.

## Figures and Tables

**Figure 1 microorganisms-13-00350-f001:**
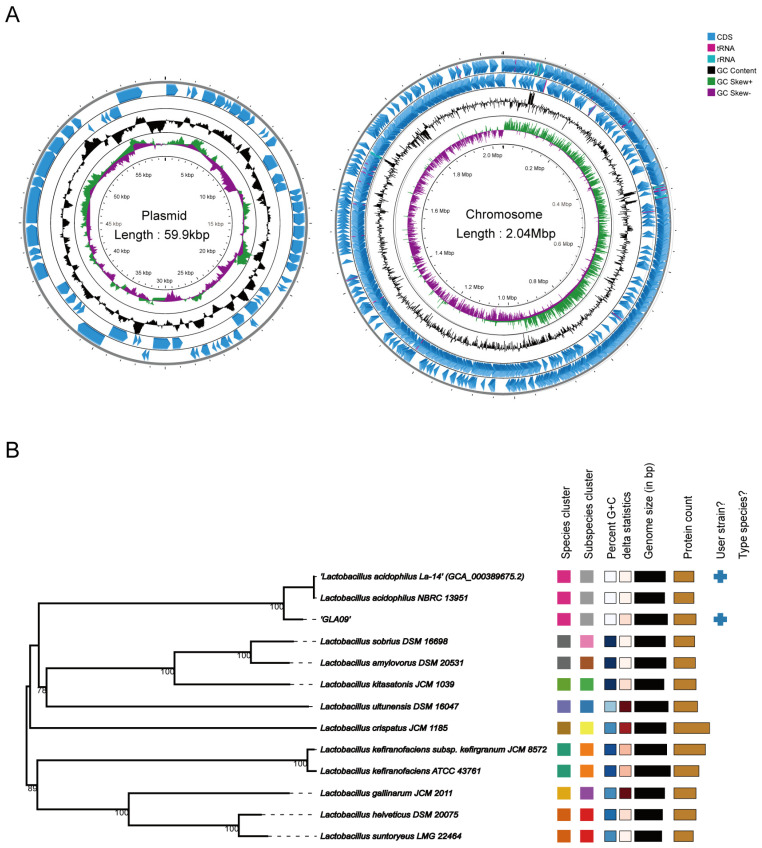
Genome analysis of *L. acidophilus* GLA09. (**A**) Genome map (circles are indicated from outside to inside: circles 1 and 2 (blue) indicate forward and reverse strands, which represent genes for CDS, tRNA, and rRNA. Circle 3 (black) indicates the GC percentage of the genome. Circle 4 (purple and green) represents GC skew). (**B**) Comparison of the genome of *L. acidophilus* GLA09 using Type genome server.

**Figure 2 microorganisms-13-00350-f002:**
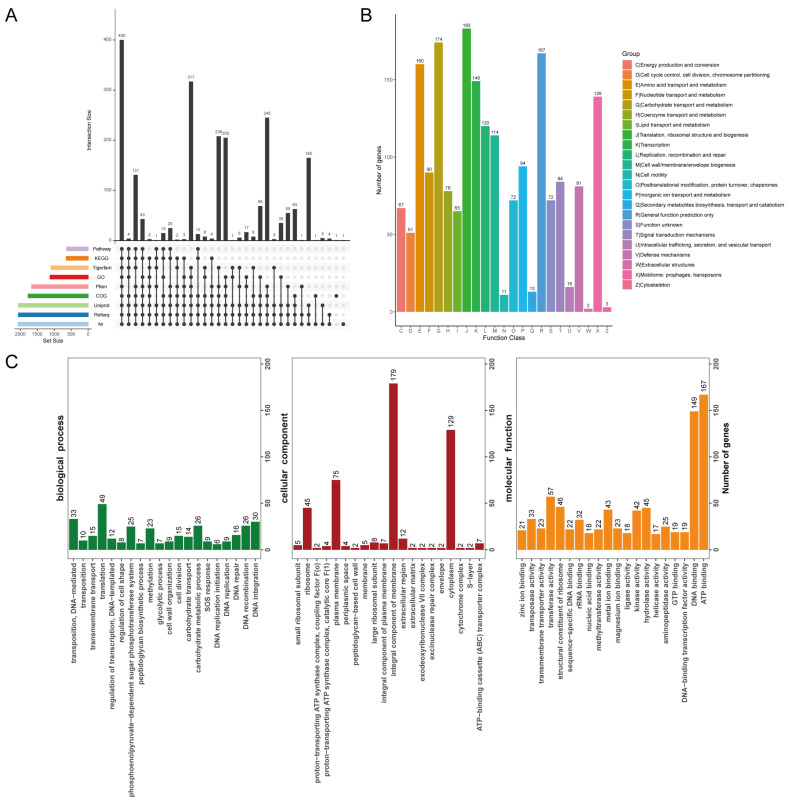
Database annotations of *L. acidophilus* GLA09. (**A**) General database annotation percentage statistics; (**B**) COG functional annotation; and (**C**) GO functional annotation of strain GLA09.

**Figure 3 microorganisms-13-00350-f003:**
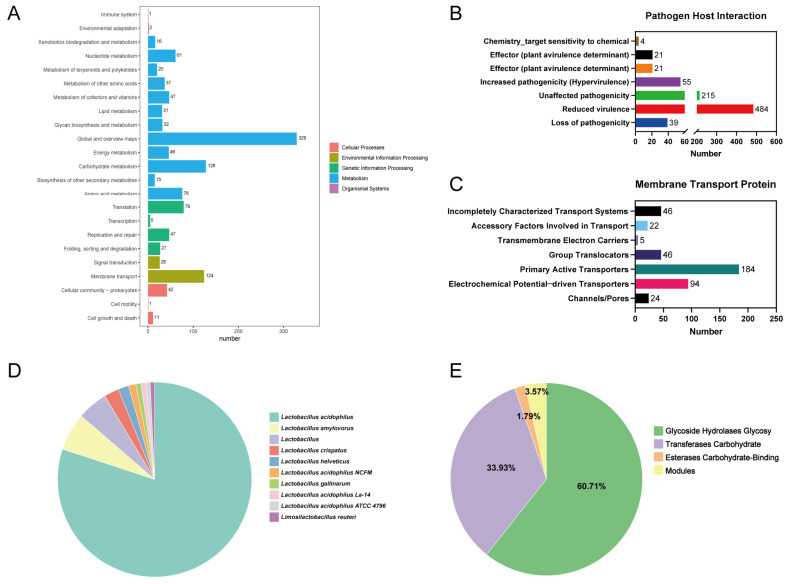
Proprietary database annotations of *L. acidophilus* GLA09. (**A**) KEGG functional annotation of *L. acidophilus* GLA09; (**B**) Pathogen Host Interactions annotations; (**C**) TCDB transporter protein; (**D**) non-redundant annotations; and (**E**) carbohydrase annotations of strain GLA09.

**Figure 4 microorganisms-13-00350-f004:**
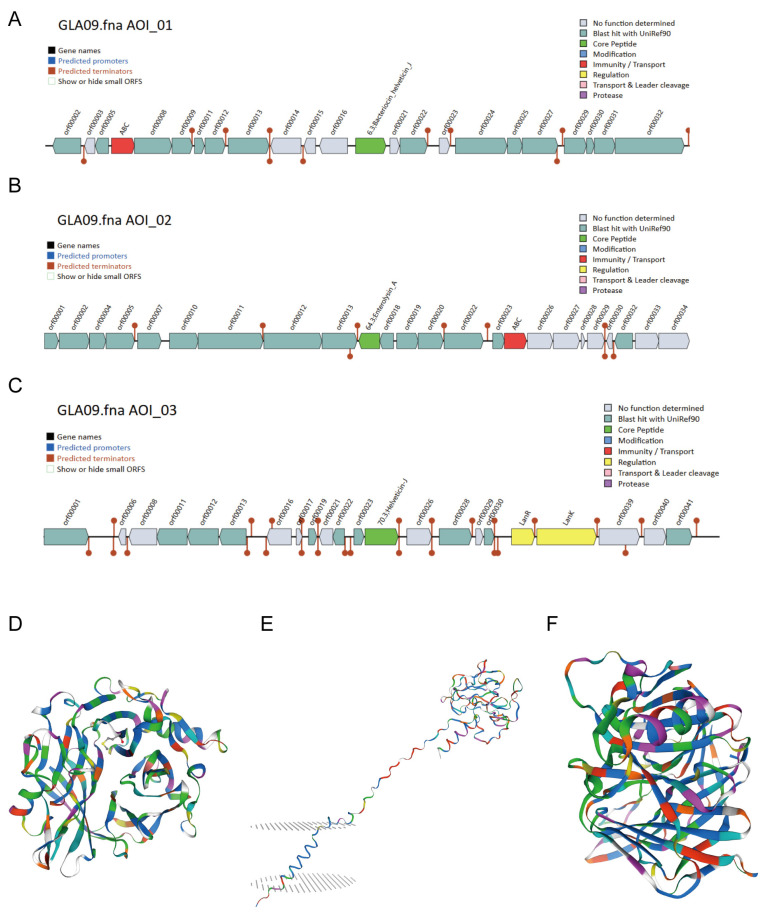
Bacteriocin prediction of *L. acidophilus* GLA09. (**A**–**C**) Cluster of genes encoding secondary metabolites and protein tertiary structure prediction of (**D**) Bacteriocin Helveticin J; (**E**) Enterolysin A; and (**F**) Helveticin J.

**Figure 5 microorganisms-13-00350-f005:**
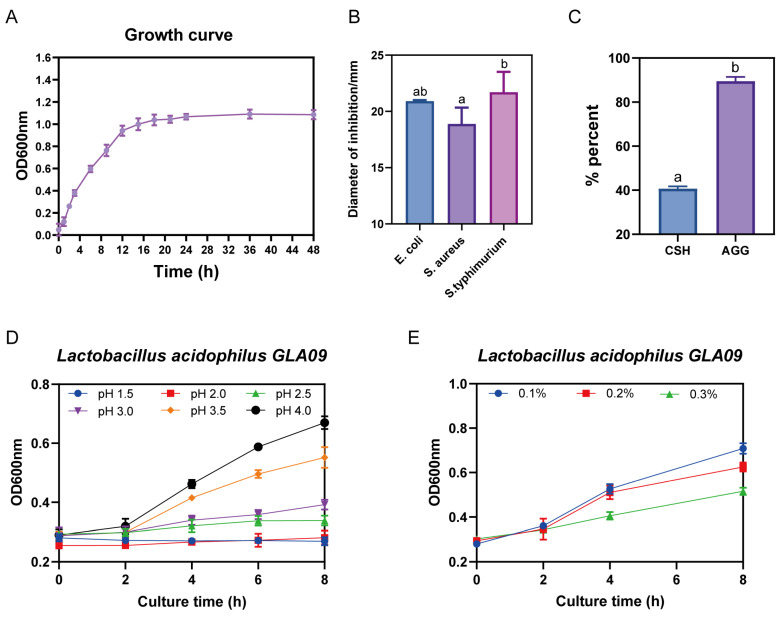
Growth curve and probiotic potential evaluation of *L. acidophilus* GLA09. (**A**) Growth curve; (**B**) in vitro bacterial inhibition; (**C**) cell surface hydrophobicity and auto-aggregation ability; (**D**) acid resistance; and (**E**) bile salt resistance. Values are displayed as the mean ± SD. The different letters in the figure represent significant differences (*p* < 0.05), the same letter in the figure indicates no significant difference (*p* > 0.05).

**Table 1 microorganisms-13-00350-t001:** Genome components of *L. acidophilus* GLA09.

Items	Values
Chromosome size (bp)	2,042,740
Plasmid size (bp)	59,983
G + C content of chromosome (%)	34.94
G + C content of plasmid (%)	33.38
tRNA	64
23S rRNA	5
16S rRNA	5
5S rRNA	5
Gene islands	8
Prophage	10

**Table 2 microorganisms-13-00350-t002:** Tolerance-related genes of *Lactobacillus acidophilus* GLA09.

Gene	Gene Name	Gene Function
Universal stress family protein		
ctg_02003	*YaaA*	peroxide stress protein
ctg_00159	-	universal stress protein
ctg_01392	-	universal stress protein
Heat stress resistance		
ctg_00883	*DnaK*	molecular chaperone DnaK
ctg_00884	*DnaJ*	molecular chaperone DnaJ
ctg_00882	*GrpE*	nucleotide exchange factor GrpE
ctg_00881	*HrcA*	heat-inducible transcriptional repressor
ctg_00290	*HslO*	Hsp33 family molecular chaperone
ctg_00427	*GroES*	co-chaperone GroES
ctg_00428	*groEL*	chaperonin GroEL
ctg_01187	*HslU*	ATP-dependent protease ATPase subunit
Cold-shock stress resistance		
ctg_01357	-	cold-shock protein
ctg_00502	*CSD*	cold-shock domain
ctg_01357	*CspA*	cold-shock protein
Bile salt resistance		
ctg_01277	*BSH*	bile salt hydrolase
ctg_01002	*ppaC*	manganese-dependent inorganic pyrophosphatase
ctg_01277	*cbh*	choloylglycine hydrolase
ctg_00059	*ldhA*	D-lactate dehydrogenase
ctg_00379	*pstB1*	phosphate import ATP-binding protein
pH stress resistance		
ctg_00826	*Asp23*	alkaline shock protein (Asp23) family
ctg_00805	*Asp23*	alkaline shock protein (Asp23) family
ctg_02131	*ClcA*	H+/Cl− antiporter ClcA
ctg_01406	*atpH*	F-type H+-transporting ATPase subunit delta
ctg_01402	*atpC*	F-type H+-transporting ATPase subunit epsilon
ctg_01405	*atpA*	F0F1 ATP synthase subunit alpha
ctg_01403	*atpD*	F-type H+/Na+-transporting ATPase subunit beta
ctg_01404	*atpG*	F-type H+-transporting ATPase subunit gamma
ctg_01408	*atpE*	F-type H+-transporting ATPase subunit c
ctg_01409	*atpB*	F-type H+-transporting ATPase subunit a
ctg_01407	*atpF*	F-type H+-transporting ATPase subunit b
ctg_01705	*nhaC*	Na+/H+ antiporter NhaC
Oxidative stress		
ctg_00443	*trxA*	thioredoxin 1
ctg_01044	-	putative NADH-flavin reductase
ctg_00417	-	NADH/NAD ratio-sensing transcriptional regulator Rex
ctg_01569	*mvaA*	hydroxymethylglutaryl-CoA reductase
ctg_01520	*trxB*, *TRR*	thioredoxin reductase (NADPH)
ctg_00566	-	thiol-disulfide isomerase or thioredoxin
ctg_01776	*pcaC*	4-carboxymuconolactone decarboxylase
ctg_00730	-	NADPH-dependent 2,4-dienoyl-CoA reductase
ctg_01536	-	OAR1`3-oxoacyl-[acyl-carrier protein] reductase
ctg_00527	-	NADH dehydrogenase
ctg_00526	-	NADH dehydrogenase
ctg_00711	-	NADH oxidase
ctg_01020	*GSR*, *gor*	glutathione reductase (NADPH)
ctg_01536	*fabG*	OAR1`3-oxoacyl-[acyl-carrier protein] reductase

**Table 3 microorganisms-13-00350-t003:** Drug resistance gene prediction of L. acidophilus GLA09.

Drug Class	Resistance Gene	Identities%	Percentage Length of Reference Sequence	AMR Gene Family
Lincosamide antibiotic	*Lnu(C)*	97.56	100	Lincosamide nucleotidyltransferase (LNU)

**Table 4 microorganisms-13-00350-t004:** Prediction of virulence factor to *Lactobacillus acidophilus* GLA09.

VFDB ID	Virulence Factor Names	Related Genes
VFG016490	*EF-Tu*	(Tuf) elongation factor Tu
VFG000077	*clpP*	(clpP) ATP-dependent Clp protease proteolytic subunit
VFG006826	*lisR*	(lisR) two-component response regulator
VFG012103	*groEL*	(groEL) chaperonin GroEL
VFG005865	*galU*	(galU) UTP--glucose-1-phosphate uridylyltransferase GalU
VFG006041	*rfbA*	(rfbA) glucose-1-phosphate thymidylyltransferase RfbA
VFG037099	*msrA/B*	(msrA/B(pilB)) trifunctional thioredoxin/methionine sulfoxide reductase A/B protein
VFG026980	*sigA*	(sigA/rpoV) RNA polymerase sigma factor

## Data Availability

The whole genome sequencing results have been uploaded to the NCBI database under the project number PRJNA1028138 (https://www.ncbi.nlm.nih.gov/bioproject/PRJNA1028138/, accessed on 14 October 2023).

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
