# Peer review of "Developing Gut-Healthy Strains for Pets: Probiotic Potential and Genomic Insights of Canine-Derived Lactobacillus acidophilus GLA09"

_microorganisms, 2025, doi:10.3390/microorganisms13020350_

Round 1
Reviewer 1 Report
Comments and Suggestions for Authors
Manuscript "Probiotic potential and genomic insights of canine-derived Lactobacillus acidophilus GLA09" shows the potential probiotic potential of this strain as well as the safety evaluated with molecular methods. I missed in the text whether the authors were looking for genes for resistance to antibiotics, which, in addition to low pathogenic potential, are an important aspect of the safe use of probiotics.
Comments on the manuscript:
line 47 - pay attention to the new nomenclature of lactobacilli
line 132- state the full name of the ATCC strain of Salmonella
I agree that the methods should be described briefly, but I think that they should be specified in more detail.
How did you adjust the concentration of bacteria in the suspension?
How did you measure OD after 2 hours when determining autoaggregation?
When measuring the growth curve, how did you solve the problem of the coloring of the broth?
Describe the antibacterial activity test in more detail and state the positive and negative control.
Fig. 2. and 3. are not transparent and should be presented better
The graphs in Fig. 5 do not show statistical significance.
Author Response
Dear Editor and Reviewer 1,
On behalf of my co-authors, we thank you very much for giving us an opportunity to revise our manuscript, and we also appreciate reviewers very much for their positive and constructive comments and suggestions on our manuscript entitled “Developing gut-healthy strains for pets: Probiotic potential and genomic insights of canine-derived Lactobacillus acidophilus GLA09” (Manuscript Number: 3451798).
We revised the manuscript according to these comments and suggestions. In general, we have tried our best to revise our manuscript and provide the point-by-point responses. All changes were marked in red in the revised manuscript. Attached please find our responses to the referees’ comments. We hope that our work can be improved again. Furthermore, we would like to show the details as follows:
Manuscript "Probiotic potential and genomic insights of canine-derived Lactobacillus acidophilus GLA09" shows the potential probiotic potential of this strain as well as the safety evaluated with molecular methods. I missed in the text whether the authors were looking for genes for resistance to antibiotics, which, in addition to low pathogenic potential, are an important aspect of the safe use of probiotics.
Response: Thanks to the reviewer's suggestion, we put the results of strain antibiotic prediction in the supplementary material (Table S2), and because of the reviewer's professional advice, we added it to the main text (Table 3) in the new version of the manuscript. In addition, the impact of antibiotic resistance on strain safety has been added to the abstract, discussion and conclusions (Line 25-26,346-356 and 403-404).
Comments on the manuscript:
Comments 1: Line 47 - pay attention to the new nomenclature of lactobacilli.
Response 1: Thank you for your professional advice, we have modified it to the new nomenclature (Line 50-51).
Comments 2. Line 132 - state the full name of the ATCC strain of Salmonella.
Response 2: Thanks to your professional opinion, we have changed the name of the Salmonella strain to Salmonella typhimurium (Line 153).
Comments 3. I agree that the methods should be described briefly, but I think that they should be specified in more detail.
Response 3: Thank you very much for your professional input, we cut the materials and methods section from our previous manuscript because of the duplication rate. We are very sorry for the trouble we have caused you with the lack of clarity. We hope that in the new version of the manuscript we have provided a clearer presentation that will allow you and the reader to better understand the materials and methods of the experiments. We've marked the changes in the text in red.
Comments 4. How did you adjust the concentration of bacteria in the suspension?
Response 4: Dear reviewers, we followed the inoculum concentration of 3.0%, the strain was incubated for 24-h, 100 uL of the bacterial solution was taken and counted by the dilution plate method, the test was repeated five times, and after taking the average value, the test results showed that the strain GLA09 reached about 5*109 CFU/mL. after that, the bacterial concentration was diluted according to the target concentration (1 × 107 CFU/mL). In the same way, the concentration of the stable phase of the pathogenic bacteria was detected, and then diluted according to the corresponding multiplicity. For each test, 100 uL of the bacterial solution was taken for dilution and plate counting to verify the concentration of the bacterial solution.
Comments 5. How did you measure OD after 2 hours when determining autoaggregation?
Response 5: The OD value was determined at 600 nm using PBS buffer as a control. We have also made detailed revisions in the manuscript to make the presentation clearer (Line 136-137).
Comments 6. When measuring the growth curve, how did you solve the problem of the coloring of the broth?
Response 6: Dear reviewers, we used MRS broth as a blank sample and then subtracted the absorbance of the experimental sample was then subtracted from the absorbance of the blank sample at the time of measurement to eliminate background effects. We have also provided a corresponding additional description in the article materials and methods (Line 81-86). Thank you for your professional opinion which makes our articles more professional and comprehensive.
Comments 7. Describe the antibacterial activity test in more detail and state the positive and negative control.
Response 7: Thanks to the reviewers for their professional advice. The negative control was blank MRS liquid medium. In addition, we set the strain previously evaluated by the laboratory as having inhibitory properties for CLP03 as a positive control (which we added to Table S3 of the supplementary material) (Line 156-157).
Table S3. Inhibition test results of L. acidophilus GLA09.
|
Items |
E. coli |
S. aureus |
S.typhimurium |
|
GLA09 |
20.91±0.09 |
18.89±1.45 |
21.71±1.81 |
|
MRS (negative control) |
- |
- |
- |
|
CLP03(positive control) |
18.74±0.80 |
17.10±1.04 |
16.09±0.42 |
Comments 8. Fig. 2. and 3. are not transparent and should be presented better.
Response 8: We are very sorry for the trouble of unclearness due to our layout problem, we have resized and uploaded the original images in the new version of the manuscript, and we hope to improve the overall clarity of the Fig.2 and Fig.3. Thanks again for your expert advice!
Comments 9. The graphs in Fig. 5 do not show statistical significance.
Response 9: Thanks to your professional opinion, we have supplemented our presentation with the results of the significance analysis in Fig.5. Different letters are used to indicate variability.
We tried our best to improve the manuscript and made some changes in the manuscript. These changes will not influence the conclusions of the paper. And here we did not list all changes but marked in red in revised paper.
We appreciate for Editors/Reviewers’ warm work earnestly, and hope that the correction will meet with approval.
Once again, thank you very much for your comments and suggestions.
Yours sincerely,
Guangyu Li
30 January 2025
College of Animal Science and Technology,
Qingdao Agricultural University, Qingdao, 266109, China.
Tel: +86-18043213500. Fax: +86-37-58957722.
E-mail:tcslgy@126.com
Reviewer 2 Report
Comments and Suggestions for Authors
The manuscript “Probiotic potential and genomic insights of canine-derived Lactobacillus acidophilus GLA09” by Zhao et al. explores the probiotic properties and genomic characteristics of Lactobacillus acidophilus GLA09, a strain isolated from healthy beagles. The study uses a combination of whole genome sequencing (WGS), in vitro analyses, and functional annotation to assess the strain's safety, gastrointestinal tolerance, and potential for industrial applications. The manuscript identifies genes responsible for tolerance to stress, bile salts, and acids, as well as genes associated with antioxidant activity and bacteriocin production. The findings suggest that GLA09 is a promising candidate for probiotic applications in the pet food industry.
Major comments
• The title accurately reflects the content but could emphasize the strain's industrial application to attract broader interest.
• The abstract provides a concise summary but could briefly mention the potential applications in the discussion.
• In Introducion the background on L. acidophilus probiotic properties is well-presented. However, a stronger emphasis on the novelty of GLA09 compared to other strains would strengthen the rationale.
• The use of bioinformatics tools (Prokka, KEGG, etc.) is standard but could benefit from a brief justification for their selection.
• The description of biogenic amine analysis (LC-MS/MS) is clear; however, the authors could elaborate on the threshold values for safety determination.
• The results are comprehensive but need a clearer comparison with other probiotic strains to highlight GLA09's unique features (genomic characterization).
• The genomic map (Figure 1) is informative but lacks sufficient labeling to explain the functional regions. Please add more details.
• Table 2 lists tolerance-related genes effectively, please add a discussion on their functional implications, to add value to the manuscript.
• Figures showcasing growth curves, acid, and bile tolerance (Figure 5) are clear but require additional replicates or statistical analysis to confirm trends (SD interpretation).
• The bacteriocin predictions (Figure 4) are intriguing but need experimental validation.
• The absence of biogenic amines is a significant finding, but the relevance of virulence factor predictions needs further elaboration.
• The discussion effectively ties genomic findings to probiotic properties but is overly focused on summarizing results. A critical analysis of potential applications and limitations is missing.
• References to industrial relevance are brief and could be expanded to include specific challenges in commercializing GLA09.
• Tables S1–S3 and Figure S1 are useful but would benefit from additional explanation in the main text. For example, Table S2 identifies a lincosamide resistance gene, but its safety implications are not discussed.
In my opinion, authors should highlight unique aspects of GLA09 compared to existing L. acidophilus strains in the introduction and discussion. If possible, I recommend incorporating in vivo studies or detailed functional assays to validate genomic predictions, particularly for bacteriocins and antioxidant activity. The ethical statement is absent, please include a clear ethical approval statement for the use of animal-derived samples. I recommend as well to clear figures and tables by labeling genomic maps more clearly to enhance interpretability and integrate supplementary materials into the main discussion to provide better context. I recommend authors discuss the broader implications of findings for industrial applications, such as scalability and regulatory challenges.
Author Response
Dear Editor and Reviewer 2,
On behalf of my co-authors, we thank you very much for giving us an opportunity to revise our manuscript, and we also appreciate reviewers very much for their positive and constructive comments and suggestions on our manuscript entitled “Developing gut-healthy strains for pets: Probiotic potential and genomic insights of canine-derived Lactobacillus acidophilus GLA09” (Manuscript Number: 3451798).
We revised the manuscript according to these comments and suggestions. In general, we have tried our best to revise our manuscript and provide the point-by-point responses. All changes were marked in red in the revised manuscript. Attached please find our responses to the referees’ comments. We hope that our work can be improved again. Furthermore, we would like to show the details as follows:
Major comments
Comments 1. The title accurately reflects the content but could emphasize the strain's industrial application to attract broader interest.
Response 1: Thank you for your professional input, we have revised the title to: Developing gut-healthy strains for pets: Probiotic potential and genomic insights of canine-derived Lactobacillus acidophilus GLA09 (Line1-4).
Comments 2. The abstract provides a concise summary but could briefly mention the potential applications in the discussion.
Response 2: Thank you for your professional input, we have added directions for future applications of the strain in the abstract. The details are as follows: In summary, L. acidophilus GLA09 has the potential to be used as a probiotic in pet food and can effectively combat intestinal health in pets (Line 29-30).
Comments 3. In Introducion the background on L. acidophilus probiotic properties is well-presented. However, a stronger emphasis on the novelty of GLA09 compared to other strains would strengthen the rationale.
Response 3: Thanks to the reviewers' comments, the novelty of strain GLA09 lies mainly in its origin from the host native gastrointestinal tract. We have added this part of the description in the article, and we thank the reviewers again for their expertise, which improved the professionalism and comprehensiveness of the article. The details are as follows: Microorganisms derived from the host are preferred as probiotics over non-host microorganisms because they have co-evolved with their hosts for approximately 10 million years, are better adapted to the host's gastrointestinal environment, exhibit superior adherence, and provide lasting effects [15–17]. Currently, while some studies on probiotics in dogs have been conducted, there remains a scarcity of research specifically focusing on canines, prompting researchers to seek improved probiotic options for the future [18]. In this study, L. acidophilus GLA09 was isolated from the gastrointestinal tract of healthy beagle dogs. The objective was to investigate the potential probiotic effects of GLA09 as a pet food probiotics and to evaluate its safety through whole genome sequencing (WGS), mass spectrometry detection, gene sequence prediction, and functional annotation, thereby providing a theoretical basis for its application (Line 55-65).
Comments 4. The use of bioinformatics tools (Prokka, KEGG, etc.) is standard but could benefit from a brief justification for their selection.
Response 4: Thanks to the reviewers' expertise, we have added the features of the databases Prokka, KEGG,GO and COG to the new version of the manuscript, i.e., the reasons for choosing these databases. The details are as follows: Prokka is a tool designed for rapid annotation of prokaryotic genomes (Line 191). The COG database is a database developed by NCBI for the annotation of homologous proteins. The strength of the COG database lies in the accuracy and comprehensiveness of its classification (202-204). GO is a database created by the Gene Ontology Consortium that provides a comprehensive description of the properties of genes and gene products in organisms (211-212). The KEGG database is a systematic analysis of the metabolic pathways of gene products in cells and the functions of these gene products (221-222).
Comments 5. The description of biogenic amine analysis (LC-MS/MS) is clear; however, the authors could elaborate on the threshold values for safety determination.
Response 5: Thanks to the reviewers for their suggestions. Although there is no specific standard for the safety value of biogenic amines, we are aware of the importance of standards, so we have referred to existing food standards. And we added the relevant standards to the article to better explain the safety of the strains (Line 379-384).
Comments 6. The results are comprehensive but need a clearer comparison with other probiotic strains to highlight GLA09's unique features (genomic characterization).
Response 6: Thanks to the reviewers' professional comments, we have added the unique features of the GLA09 genome (Line 396-405). The details are as follows: The genome mainly related to functional genes include carbohydrate metabolism, amino acid metabolism, and nucleotide metabolism and three predicted genes coding for bacteriocins. Additionally, a significant number of probiotic-related genes were identified in the L. acidophilus GLA09 genome, including trxA, mvaA, trxB, TRR, pcaC, GSR, gor, and fabG. These genes are closely associated with antioxidant functions and contribute to the unique genomic features of this strain.
Comments 7. The genomic map (Figure 1) is informative but lacks sufficient labeling to explain the functional regions. Please add more details.
Response 7: Thanks to the reviewers' comments, we have added an explanation of the functional regions of the genome map to the figure notes of Figure 1 as follows: Circles are indicated from outside to inside: circles 1 and 2 (blue) indicate forward and reverse strands, which represent genes for CDS, tRNA and rRNA. Circle 3 (black) indicates the GC percentage of the genome. Circle 4 (purple and green) represents GC skew (Line 183-186).
Comments 8. Table 2 lists tolerance-related genes effectively, please add a discussion on their functional implications, to add value to the manuscript.
Response 8: Thank you for your suggestion, we have analysed the tolerance genes of the strains and their probiotic properties in detail in our discussion (Line 357-359 and362-369).
Comments 9. Figures showcasing growth curves, acid, and bile tolerance (Figure 5) are clear but require additional replicates or statistical analysis to confirm trends (SD interpretation).
Response 9: Thanks to your professional opinion, we have supplemented our presentation with the results of the significance analysis in Fig.5.
Comments 10. The bacteriocin predictions (Figure 4) are intriguing but need experimental validation.
Response 10: We thank the reviewers for their suggestions, but we apologise that due to time constraints we are unable to add the validation test at this time. Our research primarily relies on existing data and bioinformatics analyses. The antiSMASH 7.0 method employed in our study has been effectively validated in previous research [1]. Moreover, our predictions align with data from the existing literature[2,3], further enhancing the reliability of our conclusions. We believe that the findings derived from our analyses provide a significant foundation for future studies and offer a valuable starting point for understanding the function of bacteriocins. We fully acknowledge the importance of experimental validation and plan to conduct relevant experiments in our future work to further corroborate our predictions. We appreciate your valuable guidance on our manuscript.
- Blin, K.; Shaw, S.; Augustijn, H.E.; Reitz, Z.L.; Biermann, F.; Alanjary, M.; Fetter, A.; Terlouw, B.R.; Metcalf, W.W.; Helfrich, E.J.N.; et al. antiSMASH 7.0: New and Improved Predictions for Detection, Regulation, Chemical Structures and Visualisation. Nucleic Acids Res 2023, 51, W46–W50, doi:10.1093/nar/gkad344.
- Toropov, V.A.; Shalaeva, O.N.; Roshchina, E.K.; Vakhitov, T.Y.; Sitkin, S.I. IDENTIFICATION OF BACTERIOCIN GENES IN PROBIOTIC STRAINS OF LACTIC ACID BACTERIA LACTOBACLLUS ACIDOPHILUS D-75 AND LACTOBACILLUS ACIDOPHILUS D-76. Eksp Klin Gastroenterol 2016, 58–65.
- K, D.; Sm, J.; S, J.; P, C.; N, T.; R, I.; P, D.; S, S.; V, S.; S, C. Functional and Genomic Characterization of a Novel Probiotic Lactobacillus Johnsonii KD1 against Shrimp WSSV Infection. Scientific reports 2023, 13, doi:10.1038/s41598-023-47897-w.
Comments 11. The absence of biogenic amines is a significant finding, but the relevance of virulence factor predictions needs further elaboration.
Response 11: Thanks to the reviewers' comments, we have discussed the possible role of the predicted virulence factors in more detail in the new version of the manuscript (Line 356-359 and 362-370).
Comments 12. The discussion effectively ties genomic findings to probiotic properties but is overly focused on summarizing results. A critical analysis of potential applications and limitations is missing.
Response 12: Thanks to your professional opinion, we add the limitations of the study and the possibility of future applications. The details are as follows: Although this study effectively established a connection between the genome and the probiotic properties, further validation of metabolites (bacteriocins), native animal testing, and assessments of product industrialization are necessary to provide substantial technical support for the strain as a probiotic product for pets (Line 385-391).
Comments 13. References to industrial relevance are brief and could be expanded to include specific challenges in commercializing GLA09.
Response 13: Thanks to your comments, we have added the challenges that exist in the commercialisation of the industry and the scientific issues that we need to address in the future. The details are as follows: To ensure the quality of our probiotic product, GLA09, it is essential to establish standardized production processes. This entails optimizing culture conditions, as well as handling and storage practices, to guarantee the product's effectiveness and stability (Line 388-391).
Comments 14. Tables S1–S3 and Figure S1 are useful but would benefit from additional explanation in the main text. For example, Table S2 identifies a lincosamide resistance gene, but its safety implications are not discussed.
Response 14: Thanks to your professional opinion, we focused on the antibiotic resistance gene of strain GLA09 in our discussion (Line345-355). In addition, we have added further explanatory notes on other supplementary material (Line 294-297).
Comments 15.In my opinion, authors should highlight unique aspects of GLA09 compared to existing L. acidophilus strains in the introduction and discussion. If possible, I recommend incorporating in vivo studies or detailed functional assays to validate genomic predictions, particularly for bacteriocins and antioxidant activity. The ethical statement is absent, please include a clear ethical approval statement for the use of animal-derived samples. I recommend as well to clear figures and tables by labeling genomic maps more clearly to enhance interpretability and integrate supplementary materials into the main discussion to provide better context. I recommend authors discuss the broader implications of findings for industrial applications, such as scalability and regulatory challenges.
Response 15: Thank you for your professional and comprehensive comments, which improved the readability and professionalism of the article. We have added the uniqueness of GLA09 to the new version of the manuscript and added the ethical proof (Line 422-424). We have also improved the clarity and readability of the genome map as you mentioned. And the results in the supplementary material are discussed in more detail. Finally, the technical barriers to the industrial application of the strain and the main directions for future investigations have been added to the abstract, discussion and conclusion. to its better application of strain GLA09 as a pet probiotic product. In the future, we will continue to verify the function of the strain's metabolite bacteriocin, the in vivo and ex vivo antioxidant as well as the safety test in mice, which will provide a theoretical basis for the industrial production of the strain. Thank you again!
We tried our best to improve the manuscript and made some changes in the manuscript. These changes will not influence the conclusions of the paper. And here we did not list all changes but marked in red in revised paper.
We appreciate for Editors/Reviewers’ warm work earnestly, and hope that the correction will meet with approval.
Once again, thank you very much for your comments and suggestions.
Yours sincerely,
Guangyu Li
30 January 2025
College of Animal Science and Technology,
Qingdao Agricultural University, Qingdao, 266109, China.
Tel: +86-18043213500. Fax: +86-37-58957722.
E-mail:tcslgy@126.com
Round 2
Reviewer 1 Report
Comments and Suggestions for Authors
The authors have changed everything requested and I have no further comments.
Just one small thing, and that is to write the name S. Typhimurium correctly in the graph.
Reviewer 2 Report
Comments and Suggestions for Authors
The revised manuscript responds to initial comments, and I recommend publishing it.